# Structural Characteristics of the Household Carbon Footprint in an Aging Society

Ying Long [1], Jiahao Feng [1], Aolong Sun [1], Rui Wang [2,*] and Yafei Wang [1]

1   School of Statistics, Beijing Normal University, Beijing 100875, China; 202021011005@mail.bnu.edu.cn (Y.L.); 202031011002@mail.bun.edu.cn (J.F.); 202221011005@mail.bnu.edu.cn (A.S.); ywang@bnu.edu.cn (Y.W.)
2   School of Statistics and Mathematics, Shanghai Lixin University of Accounting and Finance, Shanghai 201209, China
*   Correspondence: 18709483691@163.com

**Abstract:** The aging population has posed a challenge to China's carbon neutrality pledge. To study the household carbon footprint in an aging society, this paper has combined the age-specific consumption pattern and environmental input-output life cycle assessment (EIO-LCA) to calculate the carbon footprint of household consumption across age groups, and then identified the key pathways of carbon emissions via structural path analysis (SPA). Results indicate that the elderly contribute 11.65% to total consumption-based carbon emissions. The working group (ages 15–64) has the highest average carbon footprint (0.85 tCO$_2$e), while the elderly group (ages 65 and above) has the lowest average carbon footprint (0.82 tCO$_2$e). Urban households of all ages have a higher carbon footprint than rural households. Housing and food are the dominant sources of the elderly carbon footprint. Notably, the *production and distribution of electric power and heat power* sector associated with housing energy consumption plays a leading role in the carbon emissions pathways of elderly consumption. Measuring the carbon footprint of older people can support policy designs and decision making in key sectors along the supply chain, and further encourage low-carbon lifestyles among China's elderly. Additionally, the findings of this study have broad applications, especially for developing countries undergoing demographic transitions.

**Keywords:** carbon footprint; age-specific consumption pattern; environmental input–output life cycle assessment model; structural path analysis; the aging society

## 1. Introduction

Population aging is becoming a worldwide issue. Developed countries take the lead in entering the 'aging era' with advanced medical technology, pension policies, and low fertility rates [1]. While pursuing economic development, developing countries have continuously improved their medical infrastructure and health services, extending the life expectancy of their residents [2]. China is facing the issue of population aging [3]. According to the Seventh National Population Census, the population aged 65 years and older in the mainland reached 191 million (13.5%). By 2050, this proportion is expected to be 24.71% [4]. Changes in age structure can influence the energy demand of industrial structures and thus affect corresponding carbon emissions [5]. As one of the largest carbon emitters, China has released a series of implementation plans for carbon peaking and carbon neutrality goals. However, in the 'aging era', these mitigation policies may be challenged [6].

Household consumption is a major contributor to carbon emissions [7]. Nearly 72% of global carbon emissions are related to household consumption [8]. More than 80% of the carbon emissions in the United States can be attributed to household consumption and related economic activities [9], with 60–78% in India [10] and 30–40% in China [11,12]. Increasing carbon emissions are considered to be the main cause of global warming, which

has become one of the most urgent environmental issues. Reducing household carbon emissions can mitigate climate change and contribute to carbon neutrality targets [7]. This study focuses solely on carbon emissions caused by household consumption. The household carbon footprint is a measure of the total amount of carbon emissions that is directly and indirectly caused by household consumption activities [13]. Carbon emissions from government expenditures and the consumption of visitors are not included. Thus, the household carbon footprint in this paper mainly corresponds to the concept of the personal carbon footprint (PCF) [14].

The household carbon footprint is heterogeneous among different age groups. The lifestyles and consumption behaviors of older people are significantly distinct from those of other groups [15]. Previous research has found that the carbon footprint of older people is mainly caused by basic living needs [1]. Compared to younger people, they spend more time at home, thus consuming more residential energy [16]. In addition, the environmental preferences of older people may also be notable. Their lifestyle habits lead them to prefer low-carbon consumption patterns [17]. Existing studies need to pay more attention to the heterogeneous characteristics of the consumption behavior of people in different age groups [18], which has led to gaps in developing consumption-related emission reduction strategies.

The carbon footprint of household consumption varies among different regions. In China, the household carbon footprint shows a decreasing trend from east to west [19]. In the eastern coastal region, the developed economy supports the high consumption level of households, resulting in a higher carbon footprint. Moreover, due to the differential consumption habits of households, the energy demand intensity of products is different. For example, the heating methods of households in northern and southern regions are completely different, with the north relying on coal and the south mainly depending on electricity. These two energy sources correspond to different carbon emission intensities, further causing significant disparity in the regional distribution of the household carbon footprint. Regional energy efficiency imbalance and different consumption patterns are always considered to be the main reasons for inter-regional heterogeneity in the carbon footprint [20,21].

The household carbon footprint heterogeneity derives from differences in the consumption structure. Young and older people have differential consumption preferences and patterns [22]. As age increases, the shares of household expenditure on housing and health services tend to rise [23]. Nevertheless, older people may spend substantially less on clothing and transportation [6]. The shift in demand for consumer products affects the requirement for energy in the associated industrial sectors [9], causing a differentiated characteristic of the carbon footprint for each age cohort. Identifying the consumption patterns and carbon footprint characteristics of people in different ages can help to make age-specific mitigation strategies while meeting their living demands.

Differences in consumption patterns further shape the demand for specific industrial sectors. The production processes in industrial sectors directly generate large amounts of carbon emissions. The multiplier effect in the economic system magnifies the impact of shifts in consumption patterns on the industrial sectors [24], leading to changes in the household carbon footprint. Structural path analysis (SPA) can trace the complex interactions between sectors and decompose the pathways that significantly influence the production chain [25]. This paper identifies the key pathways and main industrial sectors of carbon emissions caused by elderly consumption via SPA. Furthermore, sectoral mitigation responsibilities are clarified, providing more accurate and detailed information for formulating policies to reduce carbon emissions in the industrial chain.

This study aims to quantify the carbon footprint of the elderly and capture their structural characteristics and key carbon emissions pathways. Based on the age-specific consumption pattern, EIO-LCA and SPA, this paper contributes to the literature as follows: (1) measuring the carbon footprint of household consumption across age groups; (2) distinguishing the characteristics of the consumption structure and carbon footprint of the

elderly; and (3) identifying the key pathways and industrial sectors that affect the carbon emissions of older people.

## 2. Literature Review

Household consumption activities are primarily responsible for carbon emissions [26]. Households contribute to carbon emissions through direct energy consumption (direct emissions) and purchases of goods and services (indirect emissions). Studies in different countries have shown that household consumption accounts for 30–80% of the national carbon emissions [9,12,27]. The Chinese economy is shifting from export-driven growth to domestic demand-driven growth [28]. At the same time, Chinese residents are steadily moving towards carbon-intensive and resource-intensive consumer lifestyles [27]. Population aging is becoming an essential social problem in China [29]. In the future, China will face the pressure of carbon reduction and imbalanced population age structures. Thus, it is necessary to pay great attention to the carbon emissions of the elderly group.

The household carbon footprint across age groups is different [30]. In some developed countries, older people are becoming the leading contributors to carbon emissions. Zheng, et al. [1] measured the GHG footprint of household consumption by different age groups in 32 developed countries. They found that senior citizens contributed up to 32.7% of national consumption-based emissions in 2015. As the most aging society in the world, Japan's older urban households have the highest per capita carbon emissions compared to other age groups [22]. Accumulated wealth and the demand for energy-intensive products, such as housing and heating, are the main reasons [5,31]. However, the literature on China draws inconsistent conclusions. Liu and Zhang [32] pointed out that Chinese seniors tend to have lower carbon emissions due to different consumption preferences. This finding corroborates with Zhang, et al. [6]. The latter revealed that urban households of all ages have a higher carbon footprint than rural households. Households with a large proportion of older people are conducive to reducing energy consumption [30], which may be attributed to their cautious and frugal attitude [33].

The impact of population aging on carbon emissions is controversial. Some studies suggest that population aging is beneficial in reducing carbon emissions [34–36]. Wang, et al. [37] found that aging can reduce environmental pressures associated with urbanization, especially in the middle- and high-income countries. On the contrary, some research argues that aging increases carbon emissions [5,38]. Fan, et al. [29] demonstrated a significant positive effect of urban population aging on carbon emissions. The demographic shift towards smaller and aging households will also contribute to environmental pressures [18]. In addition, older people may be less willing to protect the environment [39], thus generating more carbon emissions. In conclusion, the effect of demographic aging on carbon emissions needs to be clarified and further studied in depth.

Consumption represents an important vector of carbon emissions [40]. Changes in consumption can directly and indirectly affect energy use and consequently carbon emissions. Different age groups show differential preferences for energy demand intensity [32]. Zhang, et al. [6] concluded that older households require fewer industrial goods and more services than younger households. Furthermore, decreasing social activities make them spend less on clothing, entertainment, and transportation [1,41]. In comparison, they spend more time in their rooms, which consumes more residential energy [22]. Thus, some literature suggests that an aging society is strongly associated with carbon-intensive expenditure patterns [1]. Meanwhile, the demand for healthcare services by the elderly has increased substantially. Nansai, et al. [23] found that carbon emissions from patients aged 65 years and older accounted for more than half of total healthcare emissions.

Changes in consumption patterns triggered by population aging have affected the industrial sectors. SPA is often combined with input–output analysis to study carbon emissions in terms of the supply chain [42]. In economic systems, there are interdependencies among various industrial sectors [43]. SPA can divide the results of I-O analysis into different levels or pathways by extracting the links between different sectors [44]. Previous

studies have shown that industrial sectors, such as *production and distribution of electric power and heat power*, *construction*, *mining and processing of metal ores*, and *manufacture of chemical products* have higher carbon emissions [42,45,46]. Zhen, et al. [47] identified key carbon emissions pathways for urban and rural households separately, and found that *production and distribution of electric power and heat power*, *transport, storage, and postal services*, and *resident, repair and other services* are the primary sources of carbon emissions. The pathways located in the zeroth stage are the most important contributors. Moreover, inter-regional linkages play a significant role in carbon emission networks [45]. Wang, et al. [48] studied the Beijing–Tianjin–Hebei carbon emission network, and concluded that the major sectors affecting regional carbon emissions were mainly in Hebei, while Beijing had a high dependence on industrial departments of other regions. Similarly, Shanghai also has a diverse carbon emission supply chain [49].

Previous literature has mainly focused on the relationship between population aging and carbon emissions, not paying enough attention to the heterogeneity of the carbon footprint in different age groups [1,18]. Additionally, many studies use the age of household head as a representative of the entire household [6,22], masking the diversity of consumption structure among family members. This paper applied the age-specific consumption pattern to distinguish the consumption expenditure characteristics of household members at different ages. Based on EIO-LCA, the carbon footprint of household consumption across age groups is measured. The structural characteristics of the carbon footprint of the elderly are further discussed in terms of regional distribution, consumption structure and key industrial pathways, complementing the case study in developing countries facing demographic transitions.

## 3. Materials and Methods

### 3.1. Accounting Framework

The carbon footprint of household consumption can be divided into direct emissions and indirect emissions. Direct emissions are derived from household direct energy consumption, such as vehicle fuel. Indirect emissions are generated from the products and services consumed by households to meet their daily needs. Figure 1 presents the research framework of this study. The Chinese Household Income Project Survey (CHIP) database provides household expenditure information, covering eight major consumption categories: food, tobacco and liquor, clothing, residence, appliances, transport, education, healthcare and others [50,51]. Applying the age-specific consumption pattern and EIO-LCA, this paper measures the carbon footprint of households in different age groups. From the regional and urban–rural dual perspective, the article discusses the regional distribution, consumption structure and industrial pathways of the elderly carbon footprint. Simultaneously, the comparison with the children group and the working group reveals the heterogeneity of the consumption pattern of the elderly.

### 3.2. Age-Specific Consumption Pattern

Based on the age-specific consumption pattern proposed by Zhu and Wei [52], the consumption expenditure of different age groups is estimated in this paper. On the framework of the classic demand function and consumption function, this model introduces age as a dummy variable drawing on the econometric model constructed by Mankiw and Weil [53]. It can split household consumption into individuals, and the expression is as follows:

$$\begin{cases} lnE = a + blnI + c(lnI)^2 + \sum_{j=0}^{85} d_j Y_j + u \\ lnE_k = \alpha_k + \beta_k lnE + \sum_{j=0}^{85} \gamma_{k,j} Y_j + \varepsilon_k \end{cases}, (i = 1, 2, \ldots, N; \ j = 0, 5, \ldots, 85) \quad (1)$$

$E$ is the per capita expenditure on total household consumption, and $E_k$ represents the average spending on the product $k$. $I$ is the per capita net household income. $Y_j = \sum_{i=1}^{N} D_{i,j}$, indicating the number of members in the age of $j$. $D_{i,j}$ is the dummy variable and it means whether the $i$th member in the household whose age is $j$ or not. $j$ represents the age groups

with an interval of five years, and $j = 0, 5, 10, \ldots, 85$. $N$ is the number of family members. $a, b, c, d$ and $\alpha_k, \beta_k, \gamma_{k,j}$ are parameters to be estimated. $a$ and $\alpha_k$ are constant terms. $b$ is the elasticity of total household consumption with respect to income. $d_j$ represents the effect of the number of family members at age $j$ on total consumption. Similarly, $\beta_k$ denotes the elasticity of the spending on the product $k$ to total consumption. $\gamma_{k,j}$ represents the natural index of the marginal propensity to consume the $k$th product by household members in the age of $j$. $u$ and $\varepsilon_k$ are random errors.

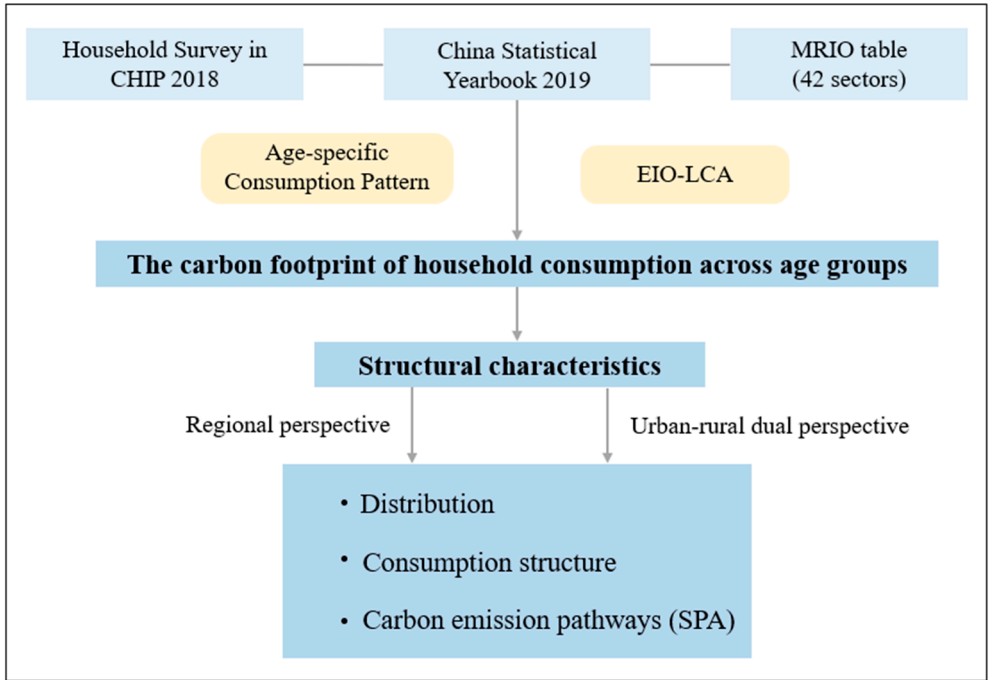

**Figure 1.** The research framework of this study.

Based on Equation (1), we can estimate the age-specific consumption expenditure:

$$\begin{cases} \widetilde{E}_j = exp(a + bln\bar{I} + c(ln\bar{I})^2 + d_j) \\ \widetilde{E}_{k,j} = exp(\alpha_k + \beta_k ln\bar{E} + \gamma_{k,j}) \end{cases} \tag{2}$$

$\widetilde{E}_j$ is the total consumption expenditure of the age group $j$, and $\widetilde{E}_{k,j}$ is the group $j$ spending on the product $k$. $\bar{I}$ and $\bar{E}$ represent the average per capita net income and per capita expenditure, respectively, in the selected sample.

Seemingly unrelated regression (SUR) was applied in Equation (1). Additionally, the estimated parameters were introduced to Equation (2) to calculate the expenditure of eight kinds of products consumed by 18 age groups in urban and rural areas.

Household expenditure data from micro-household surveys are usually lower than that of real life, thus making the results smaller [54]. The results need to be adjusted with official data from the National Bureau of Statistics (NBS). According to the information on population age distribution and per capita total consumption provided by the NBS, the adjusted per capita total consumption for three age groups, 0–14 years (children), 15–64 years (workers) and 65 years (elderly), were calculated by weighting. Then, the adjusted per capita total consumption was divided into eight consumption categories based on the proportion from Equation (2).

### 3.3. The Environmental Input–Output Life Cycle Assessment Model (EIO-LCA)

EIO-LCA quantifies the carbon footprint of household consumption of products and services by introducing environmental pressure indicators (such as carbon emissions) into the input–output framework [55]. Its fundamental purpose is to allocate direct carbon emissions from industries to final household demand. EIO-LCA bridges the macro-scale and micro-scale studies [9], and has been widely used to assess the environmental impact of household consumption [56,57].

The geographical industrial carbon emissions $Q$ are introduced into the satellite accounts of the MRIO to obtain the direct carbon emission intensity $q$ of the industrial sectors:

$$q = \frac{Q}{X} \tag{3}$$

where $X$ is the total output. $q$ represents the amount of carbon emissions produced per unit of output in a specified industry. The carbon footprint caused by household demand can be calculated by Equation (4).

$$CF = qLy = q(I - A)^{-1}y \tag{4}$$

where $CF$ represents the household carbon footprint, and $A$ is the direct consumption coefficient matrix. $q(I - A)^{-1}$ is the life-cycle carbon emission coefficient factor, which implies the total carbon footprint caused by consumption per yuan. $y$ is household consumption expenditure, including eight categories of products. A coordination matrix was established to link products and industrial sectors (Supporting Information, p. 3). We assume that the products consumed by households come from 31 provinces across the country, and the final demand matrix in the MRIO table determines the proportion of consumption in each province.

### 3.4. Structural Path Analysis (SPA)

Based on the Leontief inverse matrix, SPA explores the transmission of embodied energy paths in the economic system. $L$ can be expanded into the infinite series $L = I + A + A^2 + A^3 + \ldots = \sum_{n=0}^{\infty} A^n$. Thus, Equation (4) is equivalent to

$$\begin{aligned} CF &= q(I + A + A^2 + A^3 + \ldots + A^n + \ldots)y \\ &= \sum_{i,j=1}^{n} q_i y_{i\cdot} + \sum_{i=1}^{n} q_i \sum_{j=1}^{n} a_{i,j} y_{j\cdot} + \sum_{i=1}^{n} q_i \sum_{k=1}^{n} a_{i,k} \sum_{j=1}^{n} a_{k,j} y_{j\cdot} + \ldots, \end{aligned} \tag{5}$$

Each term $A^n$ represents the contribution of n-stage supply chains to gross output [58]. $qIy$ is the zeroth stage, representing the direct emission from the product manufacturing process. $qA^n y$ ($n > 0$) refers to the carbon emissions generated by the indirect inputs from all upstream production processes. $a_{i,j}$ is the element of $A$, $y_{j\cdot}$ is the element of $y$, and $y_{j\cdot} = \begin{cases} y_j, & \cdot = j \\ 0, & \cdot \neq j \end{cases}$. For example, $q_i a_{i,j} y_j$ represents the carbon emissions generated by sector $i$ in the first-order production layer to meet the final demand of sector $j$. Thus, we can construct a network of carbon emission paths between the final consumption and production sectors. Additionally, the number of paths at any individual s-stage is given by $N^s$. We need to identify the most significant pathways instead of finding all paths. In this paper, we present the results of the zeroth to fourth production layers, as their cumulative contribution reaches nearly 80%.

### 3.5. Data Sources

The household expenditure data and associated socio-economic backgrounds were extracted from CHIP [50,51]. It provided information on 20,000 households, including their consumption of goods and services, income, individual information, psychology, assets, and agriculture business. CHIP has been adopted by many scholars to study the consumption behavior of Chinese households in the micro field and depict the corresponding

environmental impacts [59]. The CHIP2018 sample was selected by systematic sampling method in three layers of east, center and west and contained 15 provinces. This paper further adjusts the expenditure data from CHIP based on the age structure of provincial population and total household consumption data provided by the *2019 Statistical Yearbook*, and then extrapolates the results to 30 provinces (except Tibet) reasonably.

The geographical industrial $CO_2$ emissions were calculated by Emission Coefficient Method (ECM) from IPCC2006 [60]. Terminal energy consumption in industrial subsectors was obtained from the *Provincial Statistical Yearbooks* and the *Energy Statistics Yearbook*, with the net calorific value from the *2005 Energy Statistics Yearbook*. Carbon oxidation rates and carbon emission factors for fuels in each sector came from Liu, et al. [61]. Furthermore, MRIO tables used in this paper were derived from the Chinese IElab (https://ielab.info/, accessed on 27 March 2023) in the year 2018, distinguishing each of the 31 provinces and 42 industry sectors for each province [62].

## 4. Results

### *4.1. The Household Carbon Footprint in Different Age Groups*

The average household carbon footprint was 0.84 tCO$_2$e in 2018, with 0.84 tCO$_2$e for the children group, 0.85 tCO$_2$e for the worker, and 0.82 tCO$_2$e for the elderly. In terms of province, carbon emissions generated by the elderly accounted for 7–16% of the total consumption-based carbon emissions. Older people in Liaoning, Shanghai, Sichuan and Chongqing had a higher share of carbon emissions, exceeding 14%. In contrast, carbon emissions from the elderly in Guangdong, Xinjiang and Qinghai were relatively less, which may be related to the age structure of the local population. It can be seen that there are still regional differences in the carbon footprints of the elderly.

### 4.1.1. Regional Distribution of the Household Carbon Footprint

Figure 2 presents the household carbon footprint distribution by urban and rural regions. On average, the per capita carbon footprint of urban households ranges from 0.30 to 5.05 tCO$_2$e, and that of rural households ranges from 0.11 to 1.83 tCO$_2$e. The carbon footprint of household consumption shows a decreasing trend from north to south. The northwest and north are the specific regions with high carbon emissions, both in urban and rural areas. By comparison, the household carbon footprint in the southwest is generally low.

There are apparent differences in the household carbon footprint among regions, indicating that the consumption levels and energy demand intensity in different provinces differ significantly. Inner Mongolia has the highest average household carbon footprint, reaching 3.71 tCO$_2$e. The northwest is also a high carbon emission region with the average carbon footprint of 2.25 tCO$_2$e. Household expenditures in these regions are much lower than that in Beijing, but their carbon footprints are higher instead. The reasons for this are different energy structures and efficiency [21] and household lifestyles [12]. Furthermore, households of Sichuan, Chongqing, Yunnan and Guangxi have a smaller carbon footprint for both urban and rural regions. Geographical factors and inconvenient transportation are the main causes. Due to the high altitude of the terrain, the transportation infrastructure in the southwest is relatively backward. The inconvenient transportation is not conducive to regional economic development, which lowers local households' consumption level, thus causing less carbon emissions.

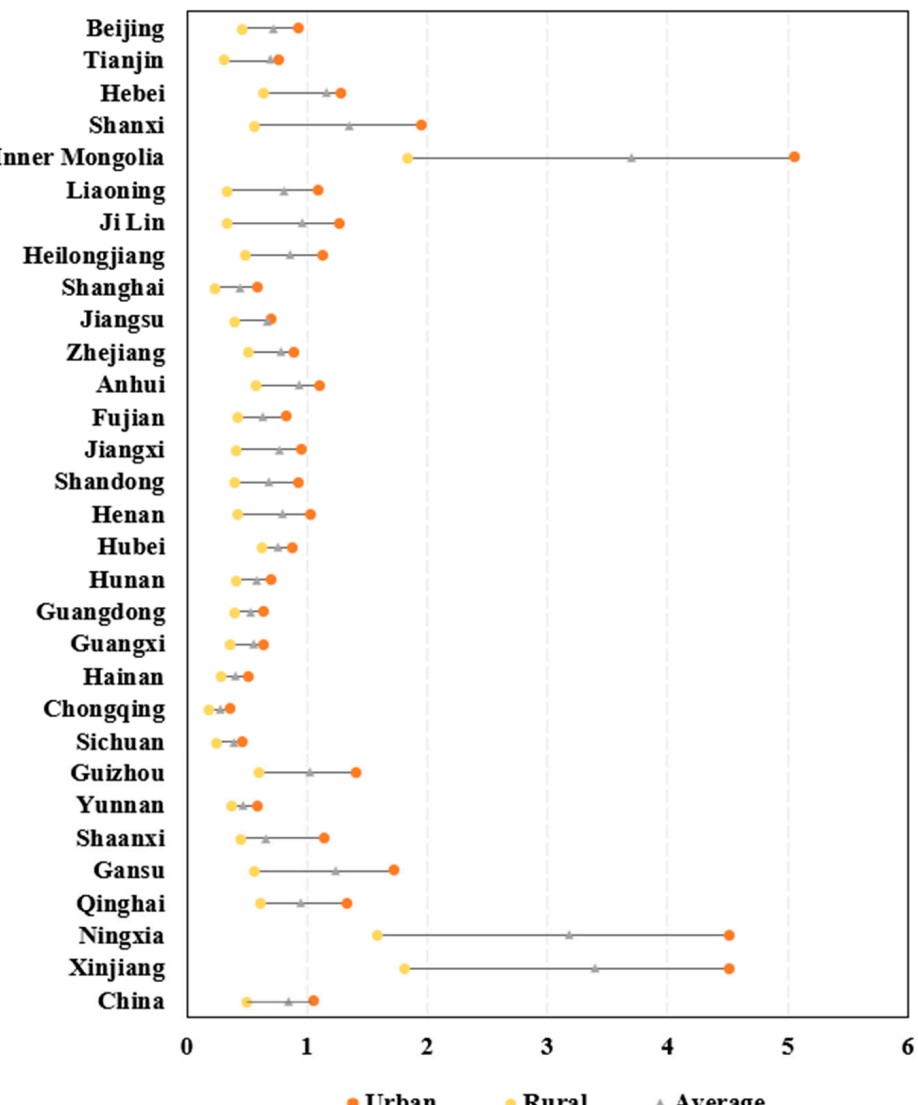

**Figure 2.** Per capita carbon footprint of household consumption in 2018 (unit: tCO$_2$e).

4.1.2. The Age-Specific Household Carbon Footprint

The household carbon footprint by age group in 30 provinces and cities is shown in Figure 3. Regarding the average carbon footprint, the working group has the most carbon emissions, while the elderly group has the lowest emissions. As the primary source of labor market, the working group acquires a large amount of labor remuneration. They are more likely to accept new consumption concepts than the elderly, and are willing to release consumption potential, resulting in a higher carbon footprint. The carbon footprint of the children group is very similar to that of the working group. In urban areas, the per capita carbon footprint of the children group is equal to that of the working group, with 1.06 tCO$_2$e. In rural regions, the carbon footprint of the working cohort is 0.49 tCO$_2$e, which is 0.01 tCO$_2$e higher than that of the children group. Children depend on their parents, the working population, for access to life resources and thus have a similar structure of consumption expenditure. Due to the decreasing occupational competitiveness, older people have less income than younger people and hence have a lower carbon footprint [63].

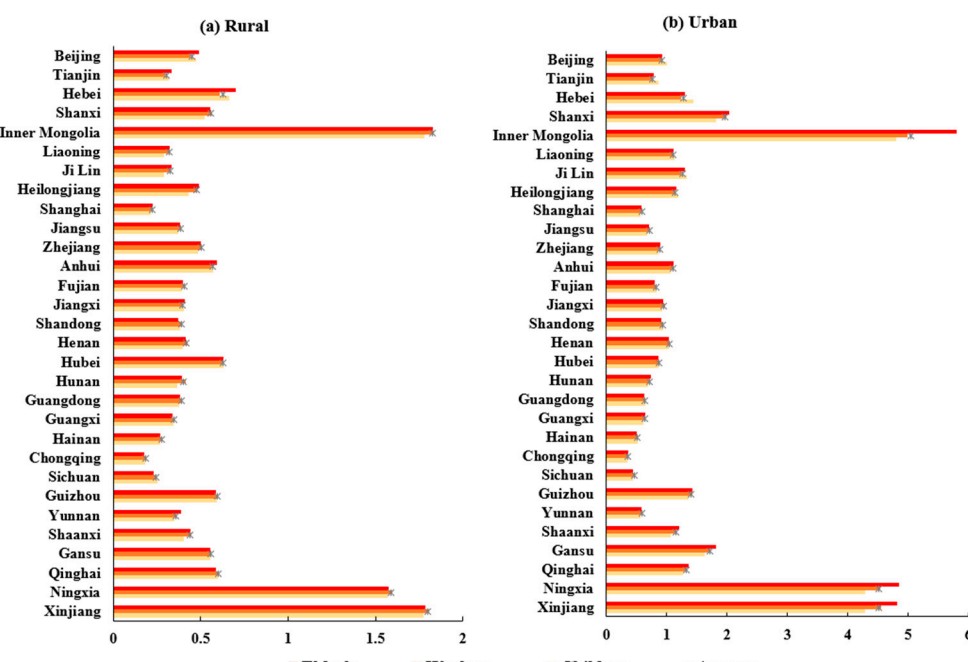

**Figure 3.** Household carbon footprint by age group in 30 provinces in 2018 (unit: tCO$_2$e).

The elderly account for 11.65% of national consumption-based carbon emissions. Although the average carbon footprint of older people is relatively low, their carbon emissions are still significant in some regions. In the Beijing–Tianjin–Hebei and Heilongjiang–Jilin–Liaoning regions, the carbon footprint of the elderly exceeds that of the working group and the children group. On the one hand, the well-developed pension system provides a stable source of income for the elderly, which can support their high consumption level [64]. On the other hand, the unique consumption structure and their preference for some carbon-intensive products generate more carbon emissions. The consumption structure of different age groups will be further discussed in Section 4.2.

The urban–rural differences in the household carbon footprint are reflected in all population age groups. For all age groups, the carbon footprint of urban households is more than twice that of rural households. The per capita carbon footprint of urban elderly reaches 1.03 tCO$_2$e, while that of rural elderly is only 0.46 tCO$_2$e. Most urban older people have pensions and other property income, whereas the rural elderly get their income from agricultural farming or government subsidies [65]. Consequently, the income of the urban elderly is significantly higher than that of the rural elderly, resulting in differences in their consumption expenditures and structures.

### 4.2. Age-Specific Consumption Structure

Household consumption structure differs among age groups (Figure 4). In this section, 30 provinces are further divided into seven major regions, as shown in Table S2 (p. 5). Regarding consumption categories, food is the most important choice for households, accounting for nearly 40% of their total consumption expenditure. The proportion of food consumption of rural households is generally higher than that of urban households. Residential expenditure is the second most preferred item. Households in the north spend approximately 30% of their total expenditure on house-related products. Moreover, housing expenditure is higher for the elderly, as they spend more time at home and use more household equipment. From the perspective of age structure, the children group has the highest percentage of education expenditure. In 2018, the proportion of education to total expenditure was 6.03% for urban children. Additionally, this ratio was only 3.36% for urban older people. Furthermore, urban households spend much more on education than rural households. For the wealthy and middle-income groups, educational expenditure is

also an investment in the labor market [66]. Working people spend more on transportation, with its share of expenditure approaching 9%. High demand for commuting generates their higher spending on transportation. Older adults have more needs for healthcare services and products than younger people. Notably, the share of healthcare expenditure of the rural elderly (6.54%) was higher than that of the urban elderly (5.95%).

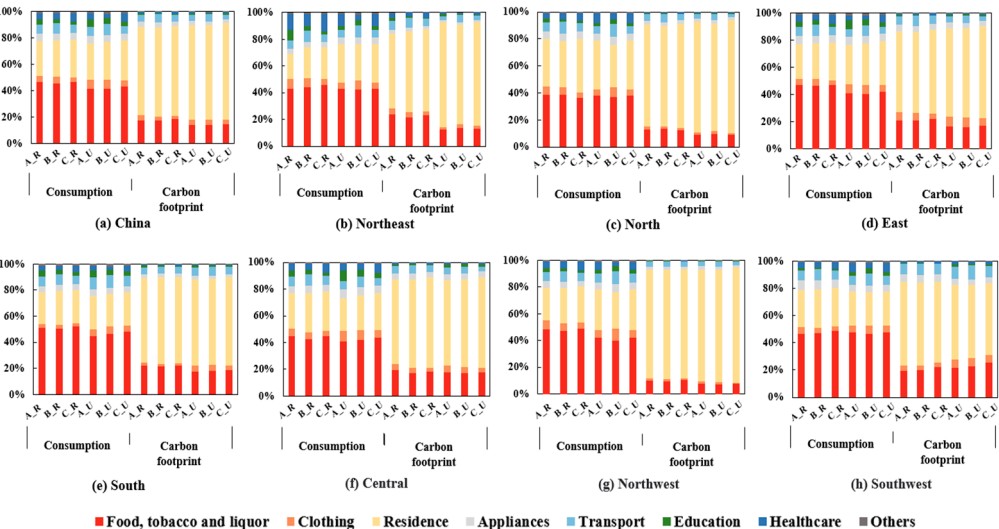

**Figure 4.** The structure of household consumption and the carbon footprint in different regions. A, B, and C represent the children group, worker group, and elderly group, respectively. R is the rural areas, and U implies the urban areas. A_R represents the rural children group.

Housing consumption is the most dominant source of the carbon footprint. The per capita carbon footprint of housing is 0.59 tCO$_2$e, nearly 70% of the household carbon footprint. There are also apparent differences in the carbon footprint of housing among regions. For example, the housing carbon footprint of urban elderly in the northwest is up to 2.16 tCO$_2$e, 7-fold higher than that of urban elderly in the southwest (0.31 tCO$_2$e). The root cause may be the differential heating methods and different energy requirements in the north and south regions. Food, the most important consumption item for households, is the second-largest source of carbon emissions (15.86%). Households in the south and southwest have a preference for food consumption, and their share of carbon emissions generated by food products exceeded 20%. Transportation is the third-largest source (5.24%). The average carbon footprint of transportation is 0.06 tCO$_2$e for the urban working group, contributing 6.49% of the total carbon footprint. For other categories of products, the carbon footprint structures are in line with the consumption structures in different ages. The carbon footprint of education expenditure is significant in the children group, while the carbon footprint of healthcare is higher in the elderly. Compared to younger people, older people have a lower demand for clothing products, leading to less associated carbon emissions. The share of the carbon footprint of appliance products does not differ significantly across age groups.

There are differences between the structure of consumption expenditure and their carbon footprint, as the carbon intensity of products differs. The carbon footprint share for housing expenditure is significantly higher than the consumption share. The reason is that the products related to housing are associated with many carbon-intensive industries, such as the metal processing and electricity power industry. This phenomenon is particularly evident in the north and northwest due to the differential energy use efficiencies and household lifestyles. Moreover, although food is the dominant product consumed by households, its carbon footprint is much lower than that of housing.

### 4.3. The Key Carbon Emission Pathways

The change in consumption patterns, combined with the multiplier effect of the economy, would further enlarge the impact of industrial sectors. This article identifies key industrial sectors and pathways of carbon emissions from elderly consumption via SPA. A threshold of 0.005% of embodied carbon emissions from the elderly is used in this study, filtering nearly 80% of the overall impact of the first four stages. The relative contributions of the pathways for each stage are shown in Table 1. It can be seen that the paths in the zeroth stage generate the most carbon emissions in the supply chain, accounting for approximately 60% of total carbon emissions from the elderly. Nevertheless, as the stage rises, the embodied carbon emissions it causes decrease. The sum of carbon emissions in the first stage is only 23.02 MtCO$_2$e, with a reduction of 71.68%. In terms of the number of paths, stage 1 has the most carbon emission pathways, almost 1.5-fold more than stage 0. It indicates that sectoral interactions have extended to the whole economy in the first stage. From the second stage onwards, the number of effective pathways decreases significantly. Most of the carbon emissions caused by paths in high stages are below the given threshold and thus not considered in this study. For research purposes, the threshold selection is subjective, so a sensitivity analysis of the thresholds is added in the Supplementary Information (pp. 27–29) of this paper.

**Table 1.** Summary of carbon emission pathways in the first four stages.

| Stage | Number of Paths | Sum of Embodied Carbon Emissions (MtCO$_2$e) | Proportion of Total Emissions | Cumulative Percentage |
|---|---|---|---|---|
| 0 | 299 | 81.29 | 59.47% | 59.47% |
| 1 | 475 | 23.02 | 16.84% | 76.31% |
| 2 | 146 | 4.12 | 3.01% | 79.32% |
| 3 | 34 | 0.68 | 0.50% | 79.82% |
| 4 | 7 | 0.10 | 0.08% | 79.89% |

#### 4.3.1. Provincial Carbon Emission Pathways

The top five supply pathways of carbon emissions caused by the elderly in 30 provinces are shown in Table S3 (p. 6). Except for Beijing, the top five paths in each region cumulatively account for over 40% of total carbon emissions, reflecting the distribution of the carbon emission supply chain of the elderly in each province. Nevertheless, this proportion is only 21.38% in Beijing, indicating that its carbon emission supply pathways are diversified, and its dependence on resources from other provinces is relatively high. For most regions, the first five pathways are in the zeroth and first stages. For paths in the second stage, carbon emissions from other industrial sectors are generally insignificant except in the northwest, Inner Mongolia, Liaoning, Guizhou and some eastern provinces. In these regions, pathways in the second stage are merely focused on indirect production activities in the *production and distribution of electric power and heat power* sector.

The top five carbon emission pathways are from local industrial sectors, except for Beijing and Qinghai. In terms of the industrial sector, *production and distribution of electric power and heat power*, *wholesale and retail trades*, and *accommodation and catering* are major contributors to carbon emissions. Most pathways associated with these sectors make zero-order contributions. In the northeast, carbon emissions directly caused by the *production and distribution of electric power and heat power* sector account for nearly 50% of total carbon emissions. Additionally, as an important component of elderly consumption, pathways of food consumption-related sectors, such as *agriculture, forestry, animal husbandry and fishery*, *food and tobacco processing*, and *transport, storage, and postal services* also have significant effects. For example, the carbon emission path from direct production in the *food and tobacco processing* sector is the most prominent carbon emission supply pathway in Sichuan, resulting in 0.75 MtCO$_2$e, 16.49% of total carbon emissions from elderly consumption. Notably, Beijing has to import a large amount of electricity from Inner Mongolia, demonstrating its

high dependence on resources from other regions. The next subsection will further discuss carbon emission pathways of Beijing's elderly consumption.

Different industrial sectors play different roles in the carbon supply chain. The *production and distribution of electric power and heat power* sector is the most prominent source of carbon emissions from elderly consumption. On the one hand, as the main provider of residential energy, it directly produces a large amount of carbon emissions. On the other hand, it is also an essential component of the supply chain in the upstream, providing electricity for production activities in other sectors. Products from *agriculture, forestry, animal husbandry and fishery* can be used as raw material inputs for other industrial sectors (e.g., *food and tobacco processing*), in addition to being consumed by households. The *transport, storage, and postal services* sector is more closely linked to production sectors in the economic system due to its transportation function. *Food and tobacco processing* and *wholesale and retail trades* are usually directly associated with final consumption and thus belong to the downstream part of the supply chain. Their upstream sectors can be derived from *production and distribution of electric power and heat power*, *animal husbandry and fishery*, *transport, storage, and postal services*, and others. Tables S4 and S5 (pp. 13–26) complement the provincial top five carbon emissions pathways caused by the younger, which are not significantly different from those of the elderly.

### 4.3.2. Key Carbon Emissions Pathways of Beijing

As a political and economic center, Beijing depends on the industrial systems of other provinces and cities to sustain the consumer demand of local households. This section studies the key carbon emissions pathways that meet the final demands of older people in Beijing. Table 2 presents the top 20 ranking pathways, whose total carbon emissions account for 32.17% of total consumption-based carbon emissions from the elderly. Furthermore, there are 14 paths originated from other provinces, contributing 35.10% of the sum of carbon emissions embodied in the top 20 ranking paths. Additionally, the outsourced carbon emissions in Beijing are imported from 12 provinces, indicating the diversity of carbon emission supply chains.

In the concerned 20 pathways, all the outsourced carbon emissions paths come from *production and distribution of electric power and heat power*. The *production and distribution of electric power and heat power* sector in Beijing cannot meet the demand of old residents, so it needs to import significant electricity from neighboring provinces, especially Inner Mongolia, Shandong and Liaoning. Additionally, electricity from Inner Mongolia is also used as an intermediate input in downstream industrial sectors such as *accommodation and catering* in Beijing. However, the *production and distribution of electric power and heat power* sector of Inner Mongolia has a high carbon emission factor, which may increase consumption-based carbon emissions [49].

**Table 2.** Top 20 key carbon emission pathways for elderly consumption in Beijing.

| Rank | Layers | Pathways | % |
|---|---|---|---|
| 1 | 0 | Beijing production and distribution of electric power and heat power | 13.07% |
| 2 | 1 | Beijing production and distribution of electric power and heat power→Beijing production and distribution of electric power and heat power | 3.68% |
| 3 | 1 | Inner Mongolia production and distribution of electric power and heat power→Beijing production and distribution of electric power and heat power | 1.75% |
| 4 | 0 | Beijing wholesale and retail trades | 1.54% |
| 5 | 1 | Shandong production and distribution of electric power and heat power→Beijing production and distribution of electric power and heat power | 1.33% |
| 6 | 1 | Liaoning production and distribution of electric power and heat power→Beijing production and distribution of electric power and heat power | 1.11% |

**Table 2.** *Cont.*

| Rank | Layers | Pathways | % |
|------|--------|----------|---|
| 7 | 2 | Beijing production and distribution of electric power and heat power→Beijing production and distribution of electric power and heat power→Beijing production and distribution of electric power and heat power | 1.04% |
| 8 | 1 | Henan production and distribution of electric power and heat power→Beijing production and distribution of electric power and heat power | 1.03% |
| 9 | 0 | Beijing transport, storage, and postal services | 1.02% |
| 10 | 1 | Anhui production and distribution of electric power and heat power→Beijing production and distribution of electric power and heat power | 0.96% |
| 11 | 1 | Xinjiang production and distribution of electric power and heat power→Beijing production and distribution of electric power and heat power | 0.72% |
| 12 | 1 | Jiangsu production and distribution of electric power and heat power→Beijing production and distribution of electric power and heat power | 0.68% |
| 13 | 1 | Hebei production and distribution of electric power and heat power→Beijing production and distribution of electric power and heat power | 0.57% |
| 14 | 1 | Shanxi production and distribution of electric power and heat power→Beijing production and distribution of electric power and heat power | 0.56% |
| 15 | 1 | Guangdong production and distribution of electric power and heat power→Beijing production and distribution of electric power and heat power | 0.55% |
| 16 | 0 | Beijing accommodation and catering | 0.53% |
| 17 | 1 | Inner Mongolia production and distribution of electric power and heat power→Beijing accommodation and catering | 0.51% |
| 18 | 1 | Fujian production and distribution of electric power and heat power→Beijing production and distribution of electric power and heat power | 0.51% |
| 19 | 1 | Hunan production and distribution of electric power and heat power→Beijing production and distribution of electric power and heat power | 0.50% |
| 20 | 2 | Inner Mongolia production and distribution of electric power and heat power→Beijing production and distribution of electric power and heat power→Beijing production and distribution of electric power and heat power | 0.49% |

## 5. Discussion

In previous studies, older people were found to be major contributors to carbon emissions in developed countries [1,22]. This paper finds that Chinese older people have a lower carbon footprint than other age groups, which is consistent with the findings of Liu and Zhang [32] and Zhang, et al. [6]. Expenditure effect and household size effect can explain this difference. Regarding the expenditure effect, older people in developed countries tend to consume more due to affluence. Considering the consumption structure, the elderly in developed countries have higher expenditures on all products except clothing products [1]. However, Chinese seniors only spend more on food and housing products to meet their basic survival needs. Additionally, there is a difference in the travelling patterns of older people between China and developed countries. Chinese older people prefer inexpensive public transportation, resulting in a lower carbon footprint. In contrast, private motorized transport is the predominant mode for older people in some developed countries [67]. In terms of consumption habits, most of the elderly in China suffered from poverty when they were young, which led to their careful and frugal consumption attitudes [6]. Furthermore, the household size effect also affects the carbon footprint of the elderly. Due to traditional cultural differences, most Chinese seniors live with their children to share living resources and reduce expenses. Multigenerational families or families with a large proportion of older people are conducive to reducing energy consumption [30].

Although the average carbon footprint of Chinese seniors is relatively low, as the number of older people increases, the total carbon footprint will become more considerable and critical. Driven by time-use and consumption patterns, future demographics will shift towards smaller and older households, thus increasing carbon emissions [18]. It is necessary to explore the structure and characteristics of the elderly carbon footprint in order to propose targeted mitigation strategies. This study could also provide lessons for developing countries undergoing demographic transitions.

The elderly carbon footprint shows significant regional disparities. Differences in economic development, consumption structure, and carbon intensity of industrial sectors are the main reasons [12]. Provincial responsibility for carbon emissions should be implemented as soon as possible, while coordinating the development of industrial structures among regions. Meanwhile, accelerating the promotion of green production technologies can also help narrow regional carbon emission differences. Moreover, the average carbon footprint of the elderly in the northeast (Heilongjiang–Jilin–Liaoning) exceeds that of other groups. The aging demographic structure is a severe issue for the northeast, mainly due to the low birth rate and high net migration ratio [68]. According to NBS, the percentage of people aged 65 and above in this area reached 17.47% in 2021. Population aging would change household consumption preferences, which in turn impacts on carbon emissions [32].

Older people in urban areas have tremendous potential to reduce carbon emissions. This paper finds that the per capita carbon footprint of the urban elderly is much larger than that of the rural elderly, consistent with Sun, et al. [57]. The pension policy and social welfare system support the income source of urban older people and maintain their high consumption levels [64]. Furthermore, differential infrastructure construction and the heterogeneity of consumption behavior result in different carbon footprints of urban and rural elderly. For example, 'health-seeking behavior' triggered by poor sanitation conditions increases the health expenditure of rural households [69]. With the acceleration of urbanization, the number of urban older people will further grow. They may become the main force driving domestic demand, and the consequent carbon emissions will be more substantial. Compared with the elderly in rural areas, urban elderly with greater reduction potential should be the target of mitigation policies.

Housing expenditure is the most significant source of the elderly carbon footprint. Older people spend more time at home, and reduce their consumption on transportation, clothing and education. The consumption structure of the elderly is rigid, with a large demand for residential energy, which means that their lifestyles are carbon-intensive [1]. While meeting their basic living needs, environmental awareness should be strengthened for older people to reduce carbon emissions. Retrofitting housing is a practical and feasible solution, such as improving housing materials and replacing inefficient heating systems. For poor elderly groups, the government should provide appropriate subsidies to improve their energy use efficiency.

The key carbon emissions pathways are identified by SPA, with the aim of understanding the distribution of responsibility for energy savings among different sectors and stages [47]. This paper finds that the carbon emission pathways for elderly consumption are concentrated in stages 0 and 1. The main pathways in most provinces and cities come from local industrial sectors. Regarding the industrial sector, the *production and distribution of electric power and heat power* sector contributes the most carbon emissions, affecting the household carbon footprint directly and indirectly. Thus, improving energy efficiency in the electricity sector (such as promoting hydroelectric power) and optimizing industrial structure would be effective mitigation measures [70]. Furthermore, *agriculture, forestry, animal husbandry and fishery* and *food and tobacco processing* also significantly influence carbon emissions from elderly consumption. People should be encouraged to purchase low-carbon products [71] and reduce the consumption of milk and meat products to reduce unnecessary carbon emissions [32].

## 6. Conclusions

On the basis of the age-specific consumption pattern, EIO-LCA and SPA, this paper studies the structural characteristics of the elderly carbon footprint. The findings of this study should compel policy makers to understand the implications of demographic transitions for climate change. This paper proposes targeted, consumption-based mitigation strategies that provide important insights for climate policy making.

Housing renovation is feasible and essential. Older people should be encouraged to improve their housing materials and upgrade efficient heating systems. They should phase out energy-intensive household appliances in favor of energy-efficient ones. For rural seniors, reducing their reliance on coal and straw, and increasing the use of cleaner energy such as natural gas are recommended. Additionally, for older people in poverty, the government could help them by making housing retrofits or giving corresponding living subsidies to improve their energy use efficiency.

The *production and distribution of electric power and heat power* sector should be a priority for emission reductions. To decrease the carbon intensity of the industrial sectors, governments can promote renewable energy development by formulating policies, laws and regulations. Subsidies should be provided to support enterprises to introduce green technological innovation and adopt cleaner energy for production. In addition to energy-intensive industrial sectors, reducing carbon emissions must focus on the impact of linkages between upstream and downstream industries in different regions. The government should also strengthen inter-regional coordination and cooperation, optimize the rational allocation of resources, and build a low-carbon supply chain network. In order to enhance energy efficiency, enterprises are encouraged to increase their investment in renewable energy research. In the practical production process, enterprises can improve production efficiency by optimizing production processes and using energy-saving equipment, thereby reducing unnecessary carbon emissions.

While recognizing insights and contributions, there are some limitations. First, the effects of government expenditures and investment are neglected in the analysis due to accounting scope limitations. Introducing the impacts of government expenditures on household consumption is beneficial to providing a more comprehensive perspective on the household carbon footprint. Second, although our sample size in this study is considerable and household expenditure data is adjusted with official data from the NBS, the assumption conditions of the age-specific consumption pattern still have an important impact on the results. Moreover, research on the behavioral heterogeneity of elderly consumers can be added in the future. Older people in this paper have a lower carbon footprint, which contradicts previous studies on developed countries. It would be interesting to explore the impact of lifestyle characteristics on the carbon footprint of older people in different cultural contexts.

**Supplementary Materials:** The following supporting information can be downloaded at: https://www.mdpi.com/article/10.3390/su151712825/s1, Figure S1: Change in the percentage of cumulative carbon emissions by stage; Figure S2: Percentage changes in the number of paths by different thresholds; Table S1: The coordination matrix of consumer products and industrial sectors; Table S2: 7 regions and their provinces and cities; Table S3: Top five carbon-emission paths for the elderly by regions; Table S4: Top five carbon-emission paths for the children group by regions; Table S5: Top five carbon-emission paths for the working group by regions.

**Author Contributions:** Conceptualization, Y.L.; data curation, J.F. and A.S.; funding acquisition, Y.W.; investigation, A.S.; methodology, Y.L.; project administration, Y.W.; software, J.F.; supervision, R.W.; writing—original draft, Y.L.; writing—review and editing, R.W. and Y.W. All authors have read and agreed to the published version of the manuscript.

**Funding:** This work was supported by National Philosophy and Social Science Foundation of China [grant number 22VRC165].

**Institutional Review Board Statement:** Not applicable.

**Informed Consent Statement:** Not applicable.

**Data Availability Statement:** The data supporting the conclusions can be obtained from Supporting Information.

**Acknowledgments:** We are grateful to those participants in this study have been very generous with their time and assistance.

**Conflicts of Interest:** The authors declare no conflict of interest.

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
