# Peer review of "Structural Characteristics of the Household Carbon Footprint in an Aging Society"

_sustainability, doi:10.3390/su151712825_

Round 1

Reviewer 1 Report

please use present tense instead of past

Reviewer 2 Report

Very interesting idea and study.

The aims of the study should be more concise. The aims appear at several points in the text of the Introduction, please put them in one place.

Suggestion:

Line 91-94 about the structure of the study is maybe unnecessary. 

Line 97-98: what about the Scope 2 background (indirect) effects of purchased electricity and district heating?

Line 172: Did you use any software, which should be referred?

Line 180: CHIP2018 and other databeses should be referred at the first apperance etc. Such as in line 269: IPCC2006 etc.

Line 269: emission factors used in the study should be referred in more details.

Line 283: in Figure 2 maybe the global sustainable CF value should be display as well.

Line 535-536: would you refer someone study to this point?

Line 560-570 are the most important conclusions.

Maybe some words about other relevant environmental impact categories of consumptions than CF should be mentioned in the study considering LCA to show the more systemic approach of the research. That could be a point, which shows, why carbon footprint is one of the most important issue in sustainability.

Reviewer 3 Report

Title: Structural Characteristics of Household Carbon Footprint in 2 Aging Society

ID: sustainability-2531982

This study analyzes the household carbon footprint with its aging society effects. Here are my suggestions for the study:

Q1) Why aging increases the carbon footprint more should be explained on a more theoretical basis. Do older people consume more than younger people? Do the elderly consume more polluting goods?

Q2) In the introduction, the presentations about the carbon footprint should be used.

Q3) Studies examining China's carbon footprint should be utilized:

https://doi.org/10.3390/w13101387

https://www.nature.com/articles/s41893-020-0504-y

Q4) Figure 1 can be transferred in a more colorful and high quality way

Q5) You can provide a more detailed conclusion.

Q6) You should specify the calculation of the carbon footprint and the importance of the carbon footprint from the household.
